# Model non-Hermitian topological operators without skin effect: A general principle of construction

**Daniel J. Salib**[1*], **Sanjib Kumar Das**[1*] and **Bitan Roy**[1†]

**1** Department of Physics, Lehigh University, Bethlehem, Pennsylvania, 18015, USA

## Abstract

We propose a general principle of constructing non-Hermitian (NH) operators for insulating and gapless topological phases in any dimension ($d$) that over an extended NH parameter regime feature real eigenvalues and zero-energy topological boundary modes, when in particular their Hermitian counterparts are also topological. However, the topological zero modes disappear when the NH operators simultaneously accommodate real and imaginary (in periodic systems) or display complex (in systems with open boundary conditions) eigenvalues. These systems are always devoid of NH skin effects, as has also been confirmed from the scaling of the inverse participation ratio, thereby extending the realm of the bulk-boundary correspondence to NH systems in terms of solely the left or right zero-energy boundary localized eigenmodes. We showcase these general and robust outcomes for NH topological insulators in $d = 1, 2$ and $3$, encompassing their higher-order incarnations, as well as for NH topological Dirac, Weyl, and nodal-loop semimetals. Possible realizations of proposed NH topological phases in designer materials, optical lattices, and classical metamaterials are highlighted.

# 1  Introduction

Nontrivial topology and geometry of electronic wavefunctions in the bulk of quantum crystals leave signatures at the boundaries (edges, surfaces, hinges and corners) in terms of robust gapless modes therein: a phenomenon known as the bulk-boundary correspondence (BBC). It plays a prominent role in the identification of topological crystals in nature and is germane for topological insulators (TIs) [1–12], topological semimetals (TSMs) [13–18], and topological superconductors [19–24]. Broadly topological phases can be classified according to the co-dimension ($d_c$) of the associated boundary modes, where $d_c = d - d_B$ and $d$ ($d_B$) is the dimensionality of the system (boundary modes). Thus an $n$th order topological phase hosts boundary modes of $d_c = n$. For example, three-dimensional topological crystals supporting surface ($d_B = 2$), hinge ($d_B = 1$) and corner ($d_B = 0$) modes are tagged as first-order, second-order and third-order topological phases, respectively [25–37].

An attempt to extend the realm of these topological phases to open quantum materials leads to non-Hermitian (NH) operators, although their exact connection with the nature of the system-to-environment interactions thus far remains illusive. Nevertheless, desired NH operators, if simple, can in principle be engineered on optical lattices [38] and in classical metamaterials [39–51]. Typically, NH operators, introduced in the context of topological phases of matter, display the NH skin effect: an accumulation of all the left and right eigenvectors at the opposite ends of a system with open boundary conditions [52–79]. Naturally, it masks the BBC in terms of left or right eigenmodes, which nonetheless is captured by their bi-orthogonal product [62]. However, a direct experimental measurement of bi-orthogonal BBC remains challenging. Therefore, construction of NH topological operators, featuring the BBC in terms of their left or right eigenvectors and thus generically devoid of the NH skin effect, is of pressing and urgent theoretical and more crucially, experimental importance. Although NH skin effect-free models have been introduced in the context of optics, photonics, and electronics [80,81], their pertinence in the field of topological phases remains unexplored so far.

Here, we outline a general principle of constructing such NH operators for TIs and TSMs in any dimension as an extension of their Hermitian counterparts, which we explicitly exemplify for systems of dimensionality $d \leq 3$. We show that the NH operators display the BBC in terms of robust zero-energy boundary modes, when all its eigenvalues are purely real. But, the system becomes trivial when the eigenvalues are simultaneously real and imaginary (in periodic systems) or complex (with open boundary conditions). See Figs. 1-5.

## 1.1  Organization

The rest of the paper is organized as follows. In the next section (Sec. 2), we review the prominent models for TIs and TSMs in $d = 1, 2$ and $3$. In Sec. 3, we propose the general principle of constructing NH topological models devoid of any NH skin effects, and exemplify it for one-, two-, and three-dimensional topological systems. Absence of the NH skin effect is further anchored from the scaling of the inverse participation ratio (IPR) in Sec. 4. We summarize the findings, propose future directions, and highlight the (meta)material pertinence of our study in Sec. 5. Additional details of our investigation are relegated to three Appendices. Specifically, in Appendix A, we discuss the source of complex eigenvalues in systems with open boundary conditions. In Appendix B, we identify symmetry criteria for the presence and ab-

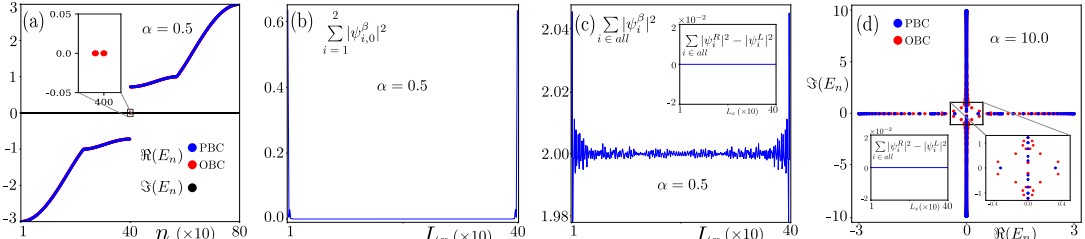

Figure 1: Non-Hermitian Su-Schrieffer-Heeger model in $d = 1$. (a) Eigenvalue spectrum for $\alpha = 0.5$ with a periodic boundary condition (PBC) and an open boundary condition (OBC), showing their guaranteed reality condition and the existence of two near zero-energy topological modes (inset) for $|\alpha| < 1$. (b) Amplitude square of the right ($\beta = R$) or left ($\beta = L$) eigenvectors of two zero-energy modes, showing their sharp localization near the ends of the chain. (c) The same as (b), but for all the right or left eigenvectors, showing no left-right asymmetry and confirming the absence of any NH skin effect (inset). (d) Eigenvalues for $\alpha = 10$, showing its generic purely real or imaginary nature (with PBC) and complex nature (with OBC), the absence of any zero-energy topological modes and skin effect for $|\alpha| > 1$. Here, we set $t = B = 1$ and $\Delta_1 = 1$. See Eqs. (2), (3) and (5), and Sec. 3.

sence of NH skin effects within a specific model in two dimensions. The topological invariant from our construction is computed for the two-dimension NH Chern insulators in Appendix C, where we also generalize it to other systems.

## 2 Topological models: A brief review

Our construction of NH topological operators is greatly facilitated by reviewing the universal model Bloch Hamiltonian for $d$-dimensional Hermitian topological phases, which can be decomposed as

$$H_{\text{Her}}(k) = H_{\text{Dir}}(k) + H_{\text{Wil}}(k) + H_{\text{HOT}}(k). \tag{1}$$

The lattice regularized Dirac kinetic energy stems from the nearest-neighbor (NN) hopping of amplitude $t$ between the orbitals of opposite parities. Explicitly, it is given by

$$H_{\text{Dir}}(k) = t \sum_{j=1}^{d} \sin(k_j a)\Gamma_j, \tag{2}$$

where $a$ is the lattice constant in a $d$-dimensional hyper-cubic lattice, momentum $k = (k_1, \cdots, k_d)$, and $k_1$, $k_2$ and $k_3$ should be identified as $k_x$, $k_y$ and $k_z$, respectively, for example. All the Hermitian $\Gamma$ matrices appearing in this work satisfy the anticommuting Clifford algebra $\{\Gamma_j, \Gamma_l\} = 2\delta_{jl}$ for any $j$ and $l$. Their dimensionality, explicit representations, and the internal structure of the associated Dirac spinor ($\Psi$) depend on the microscopic details, which we reveal while discussing specific models.

The (first-order) Wilson mass, preserving all the discrete crystal symmetries (such as, reflection, rotation and inversion), and thus transforming under the trivial singlet $A_{1g}$ representation of any crystallographic point group, is $H_{\text{Wil}}(k) = \Gamma_{d+1} m(k)$, where

$$m(k) = \Delta_1 - 2B\left[d - \sum_{j=1}^{d} \cos(k_j a)\right] + \sum_{s=1}^{p} t_s \cos(k_{d+s} a). \tag{3}$$

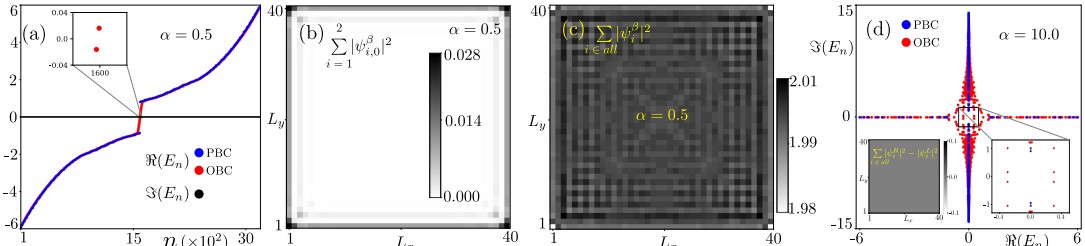

Figure 2: Non-Hermitian Qi-Wu-Zhang model in $d = 2$. (a) Eigenvalues for $\alpha = 0.5$ with PBCs and OBCs, confirming their reality condition and the existence of near zero energy topological edge modes (inset) for $|\alpha| < 1$. (b) Amplitude square of the right ($\beta = R$) or left ($\beta = L$) eigenvectors of two closest to zero energy modes, showing their sharp edge localization. (c) The same as (b), but for all the right or left eigenvectors, showing no left-right or top-bottom asymmetry about the center of the system, thus no NH skin effect. (d) Purely real or imaginary (with PBCs) and complex (with OBCs) eigenvalues for $\alpha = 10$, showing absence of any zero energy topological mode and NH skin effect for $|\alpha| > 1$. Here, we set $t = B = 1$ and $\Delta_1 = 6$. See Eqs. (2), (3) and (5), and Sec. 3.

For now we switch off the symmetry preserving out of $d$-dimensional hyperplane hopping processes by setting $t_s = 0$ for all $s$. Then the first-order Wilson mass features band inversion within the parameter regime $0 < \Delta_1/B < 4d$, where $H_{\text{Her}}^{\text{Ins}}(k) = H_{\text{Dir}}(k) + H_{\text{Wil}}(k)$ describes a $d$-dimensional first-order TI, hosting zero-energy gapless boundary modes of $d_c = 1$. Prominent examples are end modes of the Su-Schrieffer-Heeger insulator [82–84], edge modes of quantum anomalous [85] and spin Hall [5, 6] insulators, and surfaces states of three-dimensional strong $Z_2$ TIs [7–10]. Notice that $H_{\text{Dir}}(k)$ and thus $H_{\text{Her}}^{\text{Ins}}(k)$ also transform under the $A_{1g}$ representation.

A hierarchy of higher-order TIs is generated by the discrete symmetry breaking Wilson masses [35, 37]

$$H_{\text{HOT}}(k) = \Delta_2 \Gamma_{d+2} \, d_{x^2-y^2}(k) + \Delta_3 \Gamma_{d+3} \, d_{3z^2-r^2}(k), \tag{4}$$

where $d_{x^2-y^2}(k) = \cos(k_1 a) - \cos(k_2 a)$ and $d_{3z^2-r^2}(k) = 2\cos(k_3 a) - \cos(k_1 a) - \cos(k_2 a)$. The term proportional to $\Delta_2$ ($\Delta_3$) is pertinent only for $d \geq 2$ ($d \geq 3$). While $d_{x^2-y^2}(k)$ transforms under the singlet $B_{1g}$ representation of the tetragonal point group ($D_{4h}$) in $d = 2$, $d_{x^2-y^2}(k)$ and $d_{3z^2-r^2}(k)$ transform under the doublet $E_g$ representation of the cubic point group ($O_h$) in $d = 3$. By virtue of the anticommutation relation among all the $\Gamma$ matrices appearing in $H_{\text{Her}}(k)$, $H_{\text{HOT}}(k)$ acts as a mass for the gapless boundary modes of the first-order TIs in $d > 1$, and partially gaps them out, thereby yielding boundary modes with $d_c > 1$ and higher-order TIs. Below, we explain this mechanism.

As such, a finite $\Delta_2$ converts a parent first-order TI into a second-order TI. Specifically, in $d = 2$ it hosts four zero energy modes localized at the corners in the body diagonal directions ($k_1 = \pm k_2$) along which $d_{x^2-y^2}(k)$ vanishes [35]. But, in $d = 3$ it features four $z$-directional hinge modes and gapless surface states on the top and bottom $xy$ planes of a cubic crystal, exactly where $d_{x^2-y^2}(k)$ vanishes [29]. Subsequently, a finite $\Delta_2$ and $\Delta_3$ produce a third-order TI in $d = 3$, supporting zero modes at eight corners of a cubic crystal, placed on its body diagonals ($k_1 = \pm k_2 = \pm k_3$), only along which both $d_{x^2-y^2}(k)$ and $d_{3z^2-r^2}(k)$ vanish simultaneously [37]. One can continue this construction in $d \geq 3$ to realize the hierarchy higher-order TIs therein. However, we restrict ourselves to $d \leq 3$.

Finally, we consider the terms proportional to $t_s$ [Eq. (3)], yielding $(d + p)$-dimensional weak topological phases by stacking $d$-dimensional $n$th order TIs. Depending on the parameter

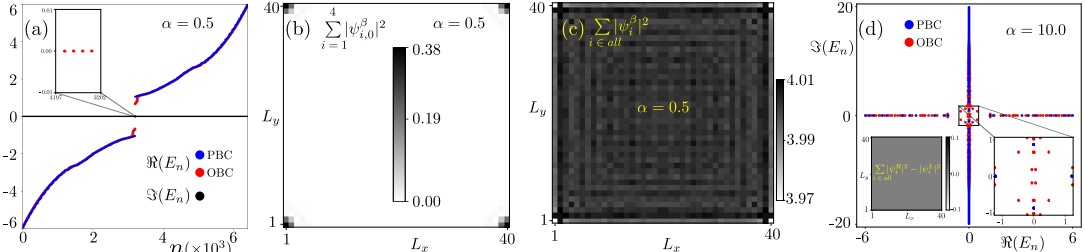

Figure 3: Non-Hermitian second-order topological insulator in $d = 2$. (a) Real eigen-value spectrum with PBCs and OBCs, accommodating four near zero-energy topological modes (inset) for $\alpha = 0.5$. (b) Amplitude square of the right ($\beta = R$) or left ($\beta = L$) eigenvectors of four closest to zero-energy modes, showing their sharp corner localization. (c) Same as (b), but for all the right or left eigenvectors, showing no left-right or top-bottom asymmetry about the center of the system, and thus no NH skin effect. (d) Generic real or imaginary (with PBCs) and complex (with OBCs) eigenvalues, and the absence of zero-energy topological modes and NH skin effect for $\alpha = 10$. Here, we set $t = B = 1$, $\Delta_1 = 6$ and $\Delta_2 = 1$. See Eqs. (2)-(5), and Sec. 3.

values ($\Delta_1/B$ and $t_s/B$), the weak topological phase can be either gapless (known as TSMs) or insulating (trivial TI). The weak phase can also be trivial, which we do not discuss here. Their gapless boundary modes appear only along the stacking direction, obtained by placing the zero-energy modes of the parent $n$th order TI in that direction. Some well known examples are the Fermi arcs of Dirac and Weyl semimetals [13–16, 86], drumhead surface states of nodal-loop semimetals [87–89], and hinge modes of higher-order Dirac semimetals [35, 90–92], which we will discuss in the context of NH TSMs. In this work, we exclusively focus on TSMs, although our results apply equally well for weak TIs. With this brief review on Hermitian TIs and TSMs, the stage is now set for us to promote their NH counterparts, devoid of any NH skin effects.

## 3   Skin effect free non-Hermitian (NH) topological operators

The key observation is that the first-order Wilson mass matrix $\Gamma_{d+1}$ anticommutes with $H_{\text{Dir}}(k)$ and $H_{\text{HOT}}(k)$. So, the products $\Gamma_{d+1}H_{\text{Dir}}(k)$ and $\Gamma_{d+1}H_{\text{HOT}}(k)$ are *anti-Hermitian*, as $(\Gamma_{d+1}\Gamma_j)^\dagger = -\Gamma_{d+1}\Gamma_j$ for $j = 1, \cdots, d, d+2, d+3$. We therefore define a NH generalization of all the topological phases in terms of the NH operator

$$H_{\text{NH}}(k, \alpha) = H_{\text{Her}}(k) + \alpha\, \Gamma_{d+1}\left[H_{\text{Dir}}(k) + H_{\text{HOT}}(k)\right]. \tag{5}$$

The parameter $\alpha$ quantifies the strength of the non-Hermiticity. Since all the matrices in $H_{\text{NH}}(k, \alpha)$ are mutually anticommuting, its eigenvalues are $\pm E_{\text{NH}}(k, \alpha)$, where

$$E_{\text{NH}}(k, \alpha) = \left[(1-\alpha^2)\left\{t^2\sum_{i=1}^{d}\sin^2(k_j a) + \Delta_2^2 d_{x^2-y^2}^2(k) + \Delta_3^2 d_{3z^2-r^2}^2(k)\right\} + m^2(k)\right]^{1/2}. \tag{6}$$

For $\alpha = 0$, we recover the energy spectra of the Hermitian systems. For $|\alpha| < 1$ all the eigen-values are purely real, showing a line gap, irrespective of the real space boundary conditions (periodic or open). For $|\alpha| > 1$ they are either purely real or imaginary, which we also find

in systems with periodic boundary conditions. However, in systems with open boundary conditions, the eigenvalues of $H_{\mathrm{NH}}(k, \alpha)$ are generically complex, as in such systems the Fourier transformed finite matrices for $\sin(k_j a)$ and $\cos(k_j a)$ do not commute with each other. A more detailed discussion on this issue is presented in Appendix A.

The NH operator $H_{\mathrm{NH}}(k, \alpha)$ also meets some non-spatial symmetries [65, 67, 75]. If $H_{\mathrm{Her}}(k)$ preserves the time-reversal ($\mathscr{T}$) and particle-hole ($\mathscr{C}$) symmetries [11], then $\mathscr{T} H_{\mathrm{NH}}^{\star}(k, \alpha) \mathscr{T}^{-1} = H_{\mathrm{NH}}(-k, \alpha)$ and $\mathscr{C} H_{\mathrm{NH}}^{\top}(k, \alpha) \mathscr{C}^{-1} = -H_{\mathrm{NH}}(-k, -\alpha)$. The sublattice ($S$) symmetry of $H_{\mathrm{NH}}(k, 0) \equiv H_{\mathrm{Her}}(k)$ (if it exists) translates into $S H_{\mathrm{NH}}(k, \alpha) S^{-1} = -H_{\mathrm{NH}}(k, -\alpha)$ for its NH counterpart. However, $H_{\mathrm{NH}}(k, \alpha)$ is devoid of the pseudo-Hermiticity symmetries.

Most importantly, as $\Gamma_{d+1}$ transforms under the $A_{1g}$ representation, $\Gamma_{d+1} H_{\mathrm{Dir}}(k)$ (also an $A_{1g}$ quantity) preserves all the spatial symmetries of the Hermitian system and $\Gamma_{d+1} H_{\mathrm{HOT}}(k)$ does not break any new crystal symmetry that has not been already broken in the Hermitian limit. The second condition is pertinent only for NH higher-order topological phases, as they break discrete rotational symmetry (such as four-fold or $C_4$) in the Hermitian limit, but preserves composite (such as $C_4 \mathscr{T}$) symmetries. Therefore, the eigenmodes of $H_{\mathrm{NH}}(k)$ do not show any NH skin effect, by construction. In Appendix B from an explicit example, we show that for the appearance of NH skin effect at least some discrete crystal symmetry must be broken, which in $d = 2$ is the inversion symmetry. Furthermore, as the anti-Hermitian component of $H_{\mathrm{NH}}(k)$ and the Hermitian operator $H_{\mathrm{Dir}}(k) + H_{\mathrm{HOT}}(k)$ vanish exactly at the same high symmetry time-reversal invariant momentum (TRIM) points in the Brillouin zone, the topological bound states (when present) are always pinned at zero energy, as in the Hermitian systems. Finally, we show that such topological zero-energy bound states exist only for $0 < \Delta_1/B < 4d$ and $|\alpha| < 1$, when the eigenvalues of $H_{\mathrm{NH}}(k)$ are purely real. Next, we anchor these general outcomes for various prominent models for topological phases of matter in one, two, and three dimensions, which we have reviewed for Hermitian systems.

## 3.1 NH topological insulator: One dimension

A NH Su-Schrieffer-Heeger model [82–84] in $d = 1$ can be defined by taking $\Gamma_1 = \tau_1$ and $\Gamma_2 = \tau_2$, where the Pauli matrices $\tau$ operate on the orbital degrees of freedom. It should be noted that the original Su-Schrieffer-Heeger model was expressed with $\Gamma_1 = \tau_2$ and $\Gamma_2 = \tau_3$, which can be obtained by unitarily rotating the form we proposed here by a unitary operator $U = \exp(i\tau_1 \pi/4) \exp(i\tau_2 \pi/4) \tau_1$. Since these two representations are unitarily equivalent, they yield identical topological properties. The results are shown in Fig. 1. Irrespective of the values of $\alpha$, this model never shows NH skin effect, as anticipated since the corresponding NH operator does not break any crystallographic symmetry, which we further anchor from the computation of the IPR in the next section. Analytical solutions of the topological modes at zero energy can be obtained by considering a hard-wall boundary at $x = 0$ such that $\Psi_0^R(x = 0) = 0$ in a semi-infinite system occupying the region $x \geq 0$, thus $\Psi_0^R(x \to \infty) = 0$. Here, the superscript '$R$' denotes right eigenvector. Such a mode can only be found at zero energy, explicitly given by

$$\Psi_0^R(x) = A \begin{pmatrix} 1 \\ \frac{t\alpha\lambda_+}{t\lambda_+ + \Delta_1 + B\lambda_+^2} \end{pmatrix} \sum_{\delta=\pm} [\delta \exp(-\lambda_\delta x)], \qquad (7)$$

where $A$ is the overall normalization constant, and

$$\lambda_\delta = \frac{t}{2B}\sqrt{1-\alpha^2} + \delta\sqrt{\frac{t^2}{4B^2}(1-\alpha^2) - \frac{\Delta_1}{B}}. \qquad (8)$$

Hence, zero-energy topological bound state can only be found if $|\alpha| < 1$, for which $\mathfrak{R}(\lambda_\delta) > 0$. As $\alpha \to 1$, such a mode becomes more delocalized. At $\alpha = 1$, the modes living on two opposite ends of the one-dimensional chain hybridize, and they disappear for $|\alpha| > 1$.

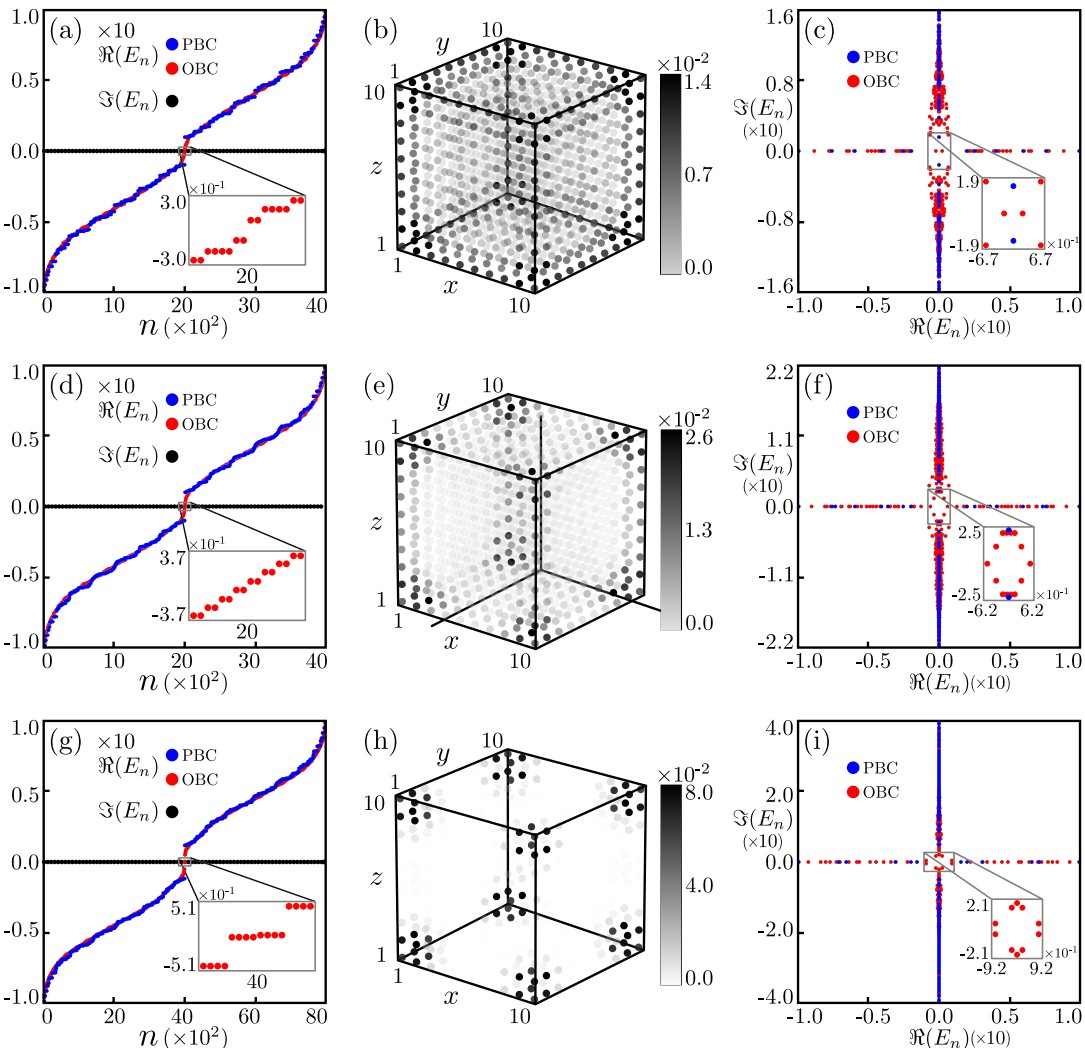

Figure 4: Hierarchy of non-Hermitian topological insulators (TIs) in $d = 3$. Real eigenvalue spectrum for $\alpha = 0.5$ with PBCs and OBCs, showing the existence of near zero-energy topological modes (insets) for (a) first-order, (d) second-order and (g) third-order TIs. Amplitude square of the right or left eigenvectors of (b) four, (e) four and (h) eight near zero-energy modes, showing sharp localization (b) on six surfaces, (e) on four $z$-directional hinges and two $xy$ surfaces, and (h) at eight corners. Here, we set $t = B = 1$, $\Delta_1 = 10$, $\Delta_2 = 1$ and $\Delta_3 = 1$ throughout. See Eqs. (2)-(5), and Sec. 3. Panels (c), (f), and (i) show purely real or imaginary (with PBCs) and complex (with OBCs) eigenvalue spectra and the absence of near-zero-energy modes in the spectrum of NH operator as in panels (a), (d), and (g), respectively, but for $\alpha = 10$.

The topological modes for the first-order TIs in $d > 1$ are obtained as the zero-energy bound states with a hard-wall boundary condition in a direction along which the translational symmetry is broken, following the steps outlined above. Subsequently, their dispersive nature is revealed by computing the matrix elements of the remaining part of the Hamiltonian, with conserved momentum in the orthogonal direction(s), within the subspace of the zero modes [12]. In higher-order TIs, topological modes of reduced dimensionality are realized by partially gapping the ones for the first-order TIs by the discrete symmetry breaking Wilson mass(es) [35, 37], as discussed in Sec. 2 for Hermitian systems. See Eq. (4). This procedure

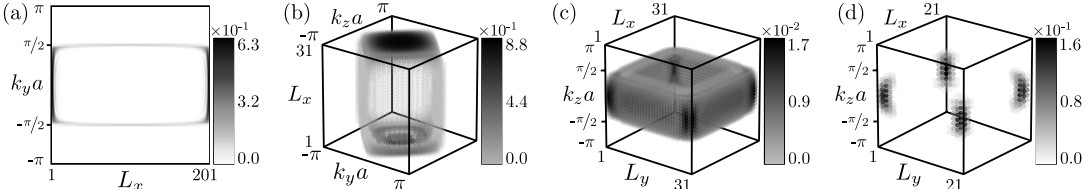

Figure 5: Non-Hermitian topological semimetals. Boundary modes of (a) a two-dimensional NH Dirac semimetal, featuring Fermi arcs between two Dirac points at $k_y = \pm\pi/(2a)$ for $\Delta_1 = 0$ and $t_2 = B = 1$, (b) a three-dimensional NH nodal-loop semimetal, showing drumhead surface states for $\Delta_1 = 0$ and $t_2 = t_3 = B = 1$, with images of the bulk nodal ring determined by $\cos(k_y a) + \cos(k_z a) = 0$ on the top and bottom surfaces, (c) a three-dimensional NH Weyl semimetal, displaying Fermi arcs in between the Weyl nodes at $k_z = \pm\pi/(2a)$, for $\Delta_1 = 0$ and $t_3 = B = 1$, and (d) a three-dimensional NH second-order Dirac semimetal with hinge Fermi arcs between two Dirac points at $k_z = \pm\pi/(2a)$ for $\Delta_1 = 0$ and $\Delta_2 = t_3 = B = 1$. Here, we set $\alpha = 0.5$. See Eqs. (2)-(5), and Sec. 3. Results are obtained from the local density of states (probability amplitude) of the right topological eigenstates. We arrive at the same conclusion from the left topological eigenstates (not shown explicitly). These findings show that the conventional bulk-boundary correspondence is operative for the NH gapless topological phases in terms of the left or right topological modes in our general principle of constructing NH skin effect-free operators, see Sec. 3.4.

applies to all the NH TIs within our construction. Hence, the above exercise proves that topological modes in NH TIs of any order in any dimension can only be found at zero energy and when $|\alpha| < 1$. So, we only provide its numerical evidence for all the remaining cases.

## 3.2 NH topological insulators: Two dimensions

A NH generalization of the Qi-Wu-Zhang model [85], describing NH Chern insulators, is realized for $\Gamma_j = \tau_j$ for $j = 1, 2, 3$. The results are shown in Fig. 2. Within the topological regime, this model hosts chiral edge modes. A NH version of the Bernevig-Hughes-Zhang model [6] for a NH quantum spin Hall insulator is realized for $\Gamma_j = \sigma_3\tau_j$ for $j = 1, 2, 3$. The Pauli matrices $\sigma$ act on the spin indices. All the results are identical to those for the NH Chern insulator, except each mode now enjoys two-fold Kramers degeneracy due to the preserved time-reversal ($\mathcal{T}$) symmetry. So, we do not show them explicitly here. Within the topological regime, this model sustains counter-propagating helical edge modes for opposite spin projections. A NH second-order TI, featuring four zero-energy corner modes [25, 35], can be realized with the addition of the $\Delta_2$ term, accompanied by the matrix $\Gamma_4 = \sigma_1\tau_0$, to the NH quantum spin Hall insulator model. The results are shown in Fig. 3. In all the cases, we numerically verify that the NH operators never show NH skin effect for any $\alpha$.

## 3.3 NH topological insulators: Three dimensions

A three-dimensional NH first-order TI, supporting gapless surface states on all six surfaces of a cubic crystal, is obtained with $\Gamma_j = \sigma_3\tau_j$ for $j = 1, 2, 3$ and $\Gamma_4 = \sigma_1\tau_0$. A NH second-order TI, supporting four $z$-directional gapless hinge and gapless $xy$ surface modes, is realized when $\Delta_2$, accompanied by $\Gamma_5 = \sigma_2\tau_0$, is finite. A three-dimensional NH third-order TI is realized when both the terms proportional to $\Delta_2$ and $\Delta_3$ are finite. As for the Hermitian third-order TI, $H_{\text{Her}}(k)$ involves six mutually anticommuting Hermitian $\Gamma$ matrices, their minimal dimensionality is eight [37, 93]. We choose $\Gamma_j = \eta_3\sigma_3\tau_j$ for $j = 1, 2, 3$, $\Gamma_4 = \eta_3\sigma_1\tau_0$,

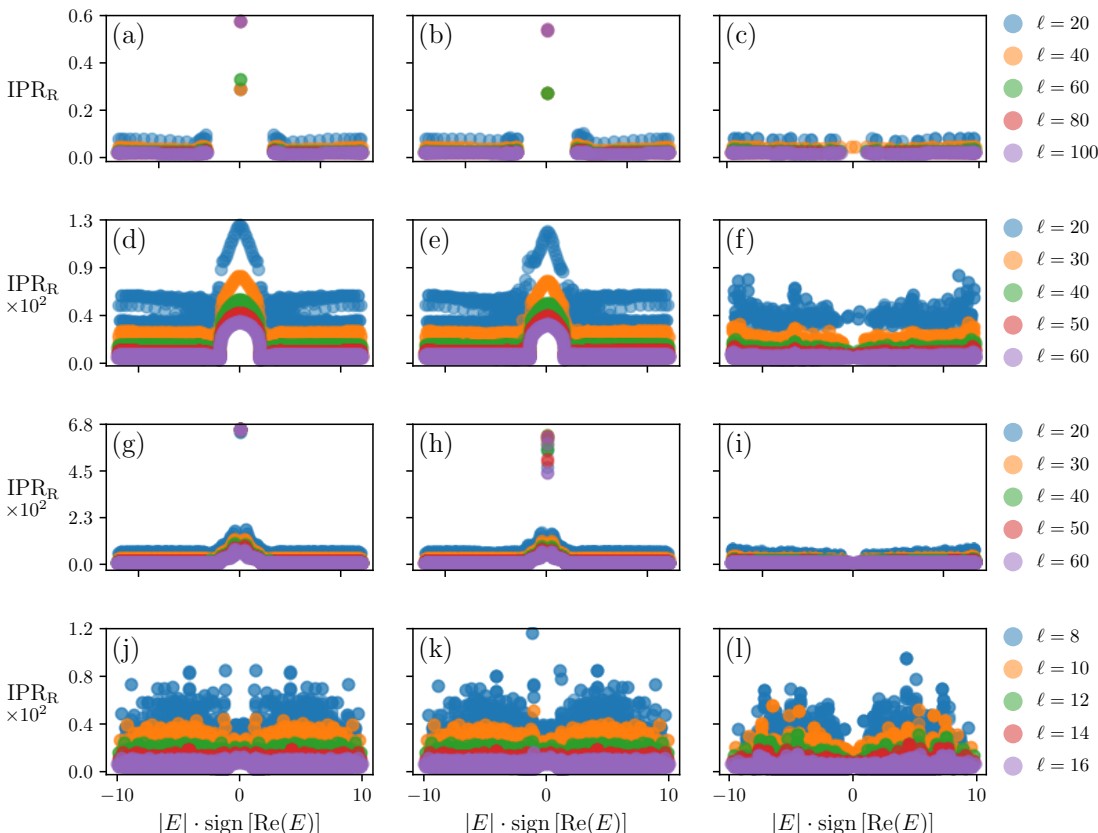

Figure 6: Inverse participation ratio (IPR) spectra of right eigenstates (IPR$_R$) for a few topological insulators discussed in Sec. 3, analyzed at different levels of non-Hermiticity ($\alpha$). The results for Hermitian systems with $\alpha = 0$ are shown in panels (a), (d), (g), and (j). The results for a moderate non-Hermiticity with $\alpha = 0.5$ where the system features non-Hermitian topology are shown in panels (b), (e), (h), and (k). Finally, the results for a strong non-Hermiticity with $\alpha = 5$ where the system is devoid of any non-trivial topology are shown in panels (c), (f), (i), and (l). The first [(a)-(c)], second [(d)-(f)], third [(g)-(i)], and fourth [(j)-(l)] rows are respectively devoted to the one-dimensional Su-Schrieffer-Heeger model for $\Delta_1 = 1$, two-dimensional Chern insulator for $\Delta_1 = 6$, two-dimensional second-order topological insulator for $\Delta_1 = 6$ and $\Delta_2 = 1$, and three-dimensional first-order topological insulator for $\Delta_1 = 10$. In the left and center column we find signatures of topological modes around zero-energy whose IPR$_R$ is higher than the rest of the extended bulk states, which are absent in the right column. But, IPR$_R$ does not show any signature of skin effects. Throughout, we take $t = B = 1$. See Sec. 4 for a detailed discussion. We find qualitatively identical results for the IPR spectra of left eigenstates (IPR$_L$).

$\Gamma_5 = \eta_3\sigma_2\tau_0$ and $\Gamma_6 = \eta_1\sigma_0\tau_0$. The set of Pauli matrices $\eta$ operate on the sublattice degrees of freedom. Then, its NH version $H_{NH}(k)$ [Eq. (5)] also supports eight zero-energy corner modes. All the results are shown in Fig. 4. In $d = 3$ as well, we numerically verified that none of NH operators display any NH skin effect (not shown explicitly, however).

## 3.4   NH Topological semimetals

By stacking NH TIs, one can realize NH TSMs depending on $\Delta_1/B$ and $t_s/B$. Naturally,

they are also devoid of any NH skin effect and support gapless boundary modes only when $|\alpha| < 1$ as their parent NH TIs. Here, we discuss some key examples. NH Su-Schrieffer-Heeger insulators, stacked in the $y$ direction, can produce a NH Dirac semimetal in $d = 2$ (like graphene) that supports Fermi arcs between two Dirac points, located along the $k_y$ axis. By continuing such stacking in the $z$ direction, we can find a NH nodal-loop semimetal, supporting drumhead surface states on the $(k_y, k_z)$ planes. By the same token, stacked (in the $z$ direction) NH Chern insulators yield a three-dimensional NH Weyl semimetal with surface Fermi arcs occupying the $(k_z, x)$ and $(k_z, y)$ planes in between two Weyl nodes along $k_z$. And stacking of NH second-order TIs produces NH higher-order Dirac semimetal in $d = 3$, featuring only $z$ directional hinge modes localized within the Dirac nodes on the $k_z$ axis. These outcomes for specific choices of $\Delta_1/B$ and $t_s/B$ are shown in Fig. 5. Notice that topological boundary modes from opposite ends of the system become connected via bulk nodal points or loops in all the NH TSMs, where the localization length of the zero-modes of the underlying NH TIs diverges. Weak TIs obtained in the same way are also devoid of any NH skin effect and accommodate topological boundary modes, which, however, occupy the entire Brillouin zone along the stacking direction(s).

## 4 Inverse participation ratio

In this section, we further anchor the absence of the NH skin effect for the models discussed in Sec. 3 from the scaling of the inverse participation ratio (IPR) with the linear system size ($\ell$) in each direction of the underlying hypercubic lattice in $d$ dimensions. Specifically, for the left ($\beta = L$) and right ($\beta = R$) eigenstates, containing the internal (orbital, spin, and sublattice, for example) indices ($\kappa$), we define the corresponding IPR as follows

$$\text{IPR}_\beta = \sum_r \left[ \sum_\kappa |\psi_\beta^\kappa(r)|^2 \right]^2, \tag{9}$$

where $\psi_\beta^\kappa(r)$ is the amplitude of a given left or right eigenstate with the internal degrees of freedom $\kappa$ at position $r$. Physically, the states that are localized around a few sites yield IPR values closer to one, while the states that spread uniformly across the entire lattice have small IPR values that roughly scales as $1/N$, where $N$ is the total number of lattice sites in the system. In the presence of any NH skin effect, the right- and left-eigenvector IPR spectra show pronounced spikes at bulk band edges, stemming from the skin localization around a specific boundary of the system. Due to the absence of any mid-gap topological modes therein, the signature of the NH skin effect (if it exists) is more clearly visible in the trivial parameter regime. For this reason, we study the IPR spectra in both topological and trivial parameter regimes, and we always impose the open boundary condition.

For concreteness, we focus on a few specific models in $d = 1, 2$, and $3$ from Sec. 3 and the spectrum for the IPR associated with the right eigenstates (IPR$_R$) are shown in Fig. 6. Since the results are qualitatively identical for the left eigenstates (IPR$_L$), we do not show them explicitly in Fig. 6. For $|\alpha| < 1$, clear signs of topological boundary mode localization persist, which is qualitatively identical to the outcomes in Hermitian systems ($\alpha = 0$). In the Su-Schrieffer-Heeger chain, we see sharp zero-energy peaks representing endpoint modes, while the 2D first-order topological insulator displays a continuous mid-gap plateau. With the 2D second-order insulator, distinct peaks at zero energy show strongly localized corner modes on top of a lower, broad plateau. In three dimensions, boundary localization for a first-order topological insulator is flattened out. Again, the IPR systematically decreases as the system size

increases in each case. Within the regime of strong non-Hermiticity ($|\alpha| = 5$), the boundary-localizations disappear completely, in agreement with our previous numerical findings and analytical computation of the topological zero-energy modes. Every state then looks similar to a typical delocalized bulk mode, distributed uniformly across the lattice with small IPR that shrinks further as the system size grows. Thus, the robust localization of boundary modes vanishes when the non-Hermitian parameters become sufficiently strong, indicating the transition into a trivial bulk regime. Most importantly, the analysis of IPR, therefore, unambiguously establishes the absence of NH skin effects in our proposed models irrespective of the parameter values, in agreement with our previous symmetry-based arguments for this phenomenon. In Appendix B, we contrast these findings with the ones when an underlying NH model displays NH skin effects.

## 5   Discussion & outlook

Here we unfold a general principle of constructing NH insulating and nodal topological phases in any dimension, belonging to any Altland-Zirnbauer symmetry class and featuring either first-order or higher-order topology, that is always devoid of the NH skin effects. In the topological regime, they showcase the BBC in terms of either the right or left zero-energy eigenmodes, when all the eigenvalues of the NH operators are purely real, irrespective of the boundary condition. The systems become trivial when these eigenvalues are purely real and imaginary in periodic systems or complex when we implement open boundary conditions. See Figs. 1-5. In the future, it will be interesting to search for a possible generalization of this construction, yielding NH skin effect free topological models that generically display *complex* eigenspectrum with periodic boundary conditions. We also note that non-reciprocity has been identified as the crucial ingredient for NH skin effects [52–79]. However, the model NH operators we proposed in this work, despite manifesting non-reciprocity in the hopping term, do not display any NH skin effect. Also, our construction for such a model in one dimension is identical to one of the proposals from a latter work for a parity and time-reversal symmetric NH model [94]. In order to numerically ensure the bi-orthonormality condition $\langle \Psi_i^L | \Psi_j^R \rangle = \delta_{ij}$ between the real space left ($L$) and right ($R$) eigenmodes of $H_{\text{NH}}(k, \alpha)$ with eigenvalues $E_i$ and $E_j$, respectively, we sometimes have to add an extremely small amount of random charge disorder ($\sim (10^{-4} - 10^{-6})t$). This is a typical limitation of numerical packages (such as ones based on LAPACK), used to diagonalize NH operators. When eigenvalues are sufficiently close and thus nearly degenerate, the bi-orthonormality condition is not necessarily satisfied and in order to restore it we need to add a small disorder that lifts such a putative degeneracy. Our construction can be immediately generalized for NH crystalline topological phases, as in the Hermitian limit their universal Bloch Hamiltonian takes the form of $H_{\text{Her}}(k)$. However, they involve longer-range hopping processes (beyond NN), allowed by crystal symmetries [95–97]. In the future, it will be worthwhile extending this construction for NH topological superconductors (TSCs). In Hermitian Dirac systems (insulating or gapless), TSCs can often be realized from local or on-site or momentum-independent pairing upon doping them, which produces a Fermi surface conducive for pairing at weak coupling [2, 98–104]. In the future, we intend to extend the realm of such local TSCs in our proposed NH skin effect-free models. Furthermore, computing the topological invariant of each NH model is left for a future investigation. Nonetheless, such a computation for the NH Chern model in $d = 2$ is explicitly shown in Appendix C, where we also relate our NH topological models to their Hermitian counterparts via a similarity transformation. However, such a mapping holds only in the parameter regime where all the eigenvalues are guaranteed to be real ($|\alpha| < 1$). Hence, for $|\alpha| < 1$ when we

find zero-energy localized topological modes in NH models, they share the same topology as their parent Hermitian systems. The quantum critical points separating NH topological and trivial insulators are described by NH Dirac or Weyl fermions. Understanding of the stability of such NH critical points against electronic interactions [105, 106] and disorder is still in its infancy.

Simplicity of our construction for the NH topological operators should make them realizable on multiple platforms, as $H_{NH}(k, \alpha)$ involves only NN hopping amplitudes and an on-site staggered potential [Eq. (5)], as also the case in the corresponding Hermitian systems. Namely, the anti-Hermitian term in our construction of $H_{NH}(k, \alpha)$ produces hopping imbalance between the NN sites in opposite directions (yielding non-Hermiticity). For example, electronic designer materials [107–111] and optical lattices constitute a promising quantum platform where these operators and the resulting NH topological phases can be realized. On optical lattices the desired hopping imbalance between NN sites can be engineered by creating two copies of a $d$-dimensional hypercubic lattice, occupied by neutral atoms living in the ground and first excited states, and coupled via running waves, such that the latter ones undergo rapid loss. When the wavelength of the running wave is equal to the lattice spacing, a NH topological operator on optical lattice with NN hopping imbalance can be realized. This proposal generalizes the one for one-dimensional NH chain with left-right hopping imbalance [59]. Although challenging, a similar engineering can in principle be executed on designer electronic materials. Topological modes in these setups can be detected by the standard scanning tunneling spectroscopy since the proposed NH topological phases are always devoid of the NH skin effect and the BBC manifests solely in terms of the left or right eigenvectors of zero-energy modes.

Classical metamaterials, such as photonic and mechanical lattices, as well as topolectric circuits, constitute yet another set of viable avenues along which the predicted skin effect free NH topology can be experimentally displayed. On all these platforms, tunable NN hopping can be implemented, and a plethora of NH topological phases with NH skin effects has already been realized [39–51]. Lack of the NH skin effect associated with all our NH operators should allow the detection of classical topological modes in these systems using well-developed tools (already applied for Hermitian topological systems), such as the two-point pump probe spectroscopy (on photonic lattices), mechanical impedance (on mechanical lattices) and electrical impedance (on topolectric circuits). It is encouraging that following our work, a topolectric circuit realization of the proposed skin effect free NH Su-Schrieffer-Heeger model has been designed [94]. Current discussion should therefore stimulate a new surge of experimental works exploring the BBC in skin effect-free NH systems in terms of solely the left and right topological eigenmodes.

## Acknowledgments

We thank Christopher A. Leong for critical reading of the manuscript.

**Funding information** D.J.S. and B.R. were supported by NSF CAREER Grant No. DMR-2238679 of B.R. S.K.D. was supported by the Startup Grant of B.R. from Lehigh University.

**Data and code availability** All the numerical codes and data are available on Zenodo [112].

★ These authors contributed equally to this work.

† Corresponding author: bitan.roy@lehigh.edu

# A  Eigenvalues with periodic and open boundary conditions

In this Appendix, we show how complex eigenvalues generically appear in the eigenspectrum of NH operator when implemented on a $d$-dimensional hyper-cubic lattice with open boundary conditions. For simplicity and concreteness, consider a one-dimensional first-order insulator with Hamiltonian

$$H(k_x) = t \sin(k_x)\Gamma_1 + [\Delta_1 - 2B(1 - \cos(k_x))]\Gamma_2, \tag{A.1}$$

where $k_x \equiv k_1$. For now we set the lattice spacing $a = 1$. Its square is given by

$$H(k_x)^2 = \left(t^2 \sin^2(k_x) + [\Delta_1 - 2B(1 - \cos(k_x))]^2\right)\Gamma_0 - 2tB[\cos(k_x), \sin(k_x)]\Gamma_2\Gamma_1, \tag{A.2}$$

where $\Gamma_0$ is the identity matrix. The commutator in the last term vanishes when $k_x$ is a good quantum number, as is the case in systems with periodic boundary conditions. However, this is no longer the case in systems with open boundary conditions. In real space, the operators $\cos(k_x)$ and $\sin(k_x)$ can be represented by matrices, where the elements at indices $(i, j)$ correspond to the local Fourier transform between site $i$ and site $j$. Then in a periodic system, they are represented by circulant real symmetric and imaginary anti-symmetric matrices, denoted by $C_x^{\mathrm{PBC}}$ and $S_x^{\mathrm{PBC}}$, respectively, and are given by

$$C_x^{\mathrm{PBC}} = \begin{bmatrix} 0 & \frac{1}{2} & 0 & \cdots & 0 & 0 & \frac{1}{2} \\ \frac{1}{2} & 0 & \frac{1}{2} & \cdots & 0 & 0 & 0 \\ 0 & \frac{1}{2} & 0 & \ddots & \vdots & \vdots & \vdots \\ \vdots & \vdots & \ddots & \ddots & \frac{1}{2} & 0 & 0 \\ 0 & 0 & \cdots & \frac{1}{2} & 0 & \frac{1}{2} & 0 \\ \frac{1}{2} & 0 & \cdots & 0 & 0 & 0 & \frac{1}{2} \end{bmatrix} \quad \text{and} \quad S_x^{\mathrm{PBC}} = \begin{bmatrix} 0 & \frac{i}{2} & 0 & \cdots & 0 & -\frac{i}{2} \\ -\frac{i}{2} & 0 & \frac{i}{2} & \cdots & 0 & 0 \\ 0 & -\frac{i}{2} & 0 & \ddots & \vdots & \vdots \\ \vdots & \vdots & \ddots & \ddots & \frac{i}{2} & 0 \\ 0 & 0 & \cdots & -\frac{i}{2} & 0 & \frac{i}{2} \\ \frac{i}{2} & 0 & \cdots & 0 & -\frac{i}{2} & 0 \end{bmatrix}. \tag{A.3}$$

The two far off-diagonal elements in each of these matrices represent the boundary hopping between the ends of the one-dimensional lattice chain. These two matrices always commute since circulant matrices form a commutative algebra. With open boundaries, the two far diagonal elements vanish and the matrices are no longer circulant. In such a system, we represent them as $C_x^{\mathrm{OBC}} = C_x^{\mathrm{PBC}} - \delta_{C_x}$ and $S_x^{\mathrm{OBC}} = S_x^{\mathrm{PBC}} - \delta_{S_x}$, where

$$\delta_{C_x} = \begin{bmatrix} 0 & 0 & \cdots & 0 & \frac{1}{2} \\ 0 & 0 & \cdots & 0 & 0 \\ \vdots & \vdots & \ddots & \vdots & \vdots \\ 0 & 0 & \cdots & 0 & 0 \\ \frac{1}{2} & 0 & \cdots & 0 & 0 \end{bmatrix} \quad \text{and} \quad \delta_{S_x} = \begin{bmatrix} 0 & 0 & \cdots & 0 & -\frac{i}{2} \\ 0 & 0 & \cdots & 0 & 0 \\ \vdots & \vdots & \ddots & \vdots & \vdots \\ 0 & 0 & \cdots & 0 & 0 \\ \frac{i}{2} & 0 & \cdots & 0 & 0 \end{bmatrix}. \tag{A.4}$$

Then the commutator appearing in the last term of Eq. (A.2) becomes

$$\left[C_x^{\mathrm{OBC}}, S_x^{\mathrm{OBC}}\right] = \frac{i}{2} \begin{bmatrix} -1 & 0 & \cdots & 0 & 0 \\ 0 & 0 & \cdots & 0 & 0 \\ \vdots & \vdots & \ddots & \vdots & \vdots \\ 0 & 0 & \cdots & 0 & 0 \\ 0 & 0 & \cdots & 0 & 1 \end{bmatrix} \equiv \frac{i}{2}\varepsilon. \tag{A.5}$$

In higher dimensions instead of a single matrix $\varepsilon$, we find $d$ number of diagonal matrices, $\varepsilon_j$ with $j = 1, \cdots, d$. The diagonal matrix $\varepsilon_j$ appears with $-1$ at diagonal indices corresponding

| Anti-Hermitian matrix | $R_x$ | $R_y$ | $R_\pi^z$ | $R_{\pi/2}^z$ | NH skin effect |
|---|---|---|---|---|---|
| $i\tau_1$ | + | − | − | Vector ($i\tau_2$) | Yes and along the $x$ directional edges |
| $i\tau_2$ | − | + | − | Vector ($-i\tau_1$) | Yes and along the $y$ directional edges |
| $i\tau_3$ | − | − | + | Scalar | No |

Table 1: Transformations of all the local anti-Hermitian operators [Eq. (B.2)] under the reflections about the $x$ axis ($R_x$) and the $y$ axis ($R_y$), the inversion ($R_\pi^z$) and the four-fold rotation ($R_{\pi/2}^z$). The last column shows whether a particular local anti-Hermitian term produces NH skin effect or not, and in which direction. See Figs. 7-9. Here, + (−) indicates whether a local anti-Hermitian term is even (odd) under a given symmetry operation.

to one boundary of the system and $+1$ at the opposite boundary in the $j$th direction, while all other diagonal elements are zero. We can then generalize this notion to our general NH models, the square of which takes the form

$$H_{\text{NH}}^2 = (1-\alpha^2)\left(H_{\text{Dir}}^2 + H_{\text{HOT}}^2\right) + H_{\text{Wil}}^2 + V, \tag{A.6}$$

where

$$V = -tB\left(\Gamma_{d+1} + \alpha\Gamma_0\right)\sum_{j=1}^{d}\varepsilon_j i\Gamma_j. \tag{A.7}$$

with the eigenvalues

$$\lambda_V = \pm tB\sqrt{1-\alpha^2}\sum_{j=1}^{d}\delta_{\text{edge}}^j. \tag{A.8}$$

Here, $\delta_{\text{edge}}^j = 1$ for sites on the edge of the system in the $j$th direction and zero elsewhere. Then, $V$ is effectively a mass perturbation to $H_{\text{NH}}^2$ at these sites, and this perturbation is imaginary for $|\alpha| > 1$. The eigenvalues of $H_{\text{NH}}$ when open boundaries are imposed now become (to leading order)

$$E_{\text{NH}} \approx \pm\sqrt{(1-\alpha^2)\left(E_{\text{Dir}}^2 + E_{\text{HOT}}^2\right) + E_{\text{Wil}}^2 + \langle n|V|n\rangle}, \tag{A.9}$$

for a given eigenstate $|n\rangle$ of the 'unperturbed' Hamiltonian (with periodic boundary condition). Here, $E_{\text{Dir}}^2$, $E_{\text{HOT}}^2$, and $E_{\text{Wil}}^2$ are the eigenvalues of $H_{\text{Dir}}^2$, $H_{\text{HOT}}^2$, and $H_{\text{Wil}}^2$ in periodic systems. Introducing this potentially imaginary (for $|\alpha| > 1$) term under the radical leads to complex eigenvalues in systems with open boundary conditions, as opposed to purely real or purely imaginary eigenvalues in periodic systems. Evaluating the perturbation to higher orders will still always yield complex eigenvalues whenever $|\alpha| > 1$.

# B   Discrete symmetries and NH skin effects: An example

In this Appendix, by focusing on a specific example of a two-dimensional Chern insulator, we show that for the appearance of NH skin effects, the spatial inversion symmetry must be broken by the corresponding NH operator. Here we use the notation $k_x \equiv k_1$ and $k_y \equiv k_2$.

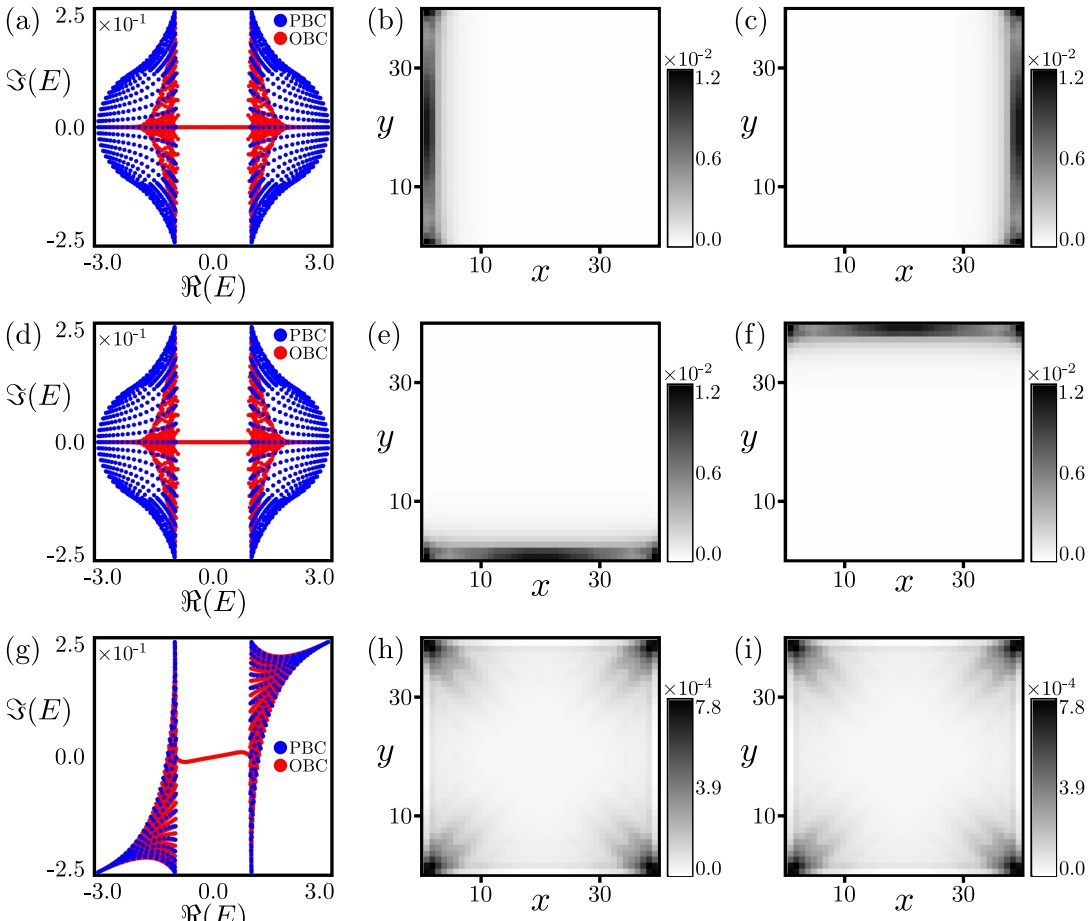

Figure 7: Eigenvalue spectra of the NH operator $H_{\mathbf{Chern}}^{\mathbf{Her}} + H_{\mathbf{AH}}(\delta_1, \delta_2, \delta_3)$ [see Eqs. (B.1) and (B.2)] for (a) $\delta_1 = 0.25$ and $\delta_2 = \delta_3 = 0$, (d) $\delta_2 = 0.25$ and $\delta_1 = \delta_3 = 0$, and (g) $\delta_3 = 0.25$ and $\delta_1 = \delta_2 = 0$. Throughout we set $\Delta_1 = 6$, $B = 1$ and $t = 1$. The amplitude square of all the left [right] eigenvector of the corresponding NH operator are shown in (b), (e) and (h) [(c), (f) and (i)], respectively. Thus, NH skin effect appears only for $\delta_1$ and $\delta_2$, respectively showing left-right and top-bottom asymmetry in the spectral weight of all the left or right eigenvectors. While for $\delta_3$ there is no left-right or top-bottom asymmetry in the spectral weight of all the left or right eigenvectors, thus yielding no NH skin effect. These outcomes are compatible with the symmetry analysis of each local anti-Hermitian term shown in Table 1. Consult Appendix B. For the chosen parameter values, the NH operator is in the topological regime as can be seen from the near zero energy modes only found in systems with open boundary conditions. However, the existence (absence) of the NH skin effect is insensitive to the topology of the model.

We begin with the square lattice model for Hermitian Chern insulator in $d = 2$, described by the Bloch Hamiltonian $H_{\mathbf{Her}}(\mathbf{k})$ [see Eqs. (1)-(4)] with $t_s = \Delta_2 = \Delta_3 = 0$ and $\Gamma_j = \tau_j$ (two-dimensional Pauli matrices) for $j = 1, 2, 3$ therein, which we explicitly write as

$$H_{\mathbf{Chern}}^{\mathbf{Her}} = t\left[\sin(k_x a)\tau_1 + \sin(k_y a)\tau_2\right] + \left[\Delta_1 - 4B + 2B\left\{\cos(k_x a) + \cos(k_y a)\right\}\right]\tau_3. \quad \text{(B.1)}$$

For the realization of the Chern insulator no spatial or crystallographic symmetry is required. However, for the discussion on the NH skin effect it is important that we identify its spatial symmetries, when this model is implemented on a square lattice via Fourier transformation.

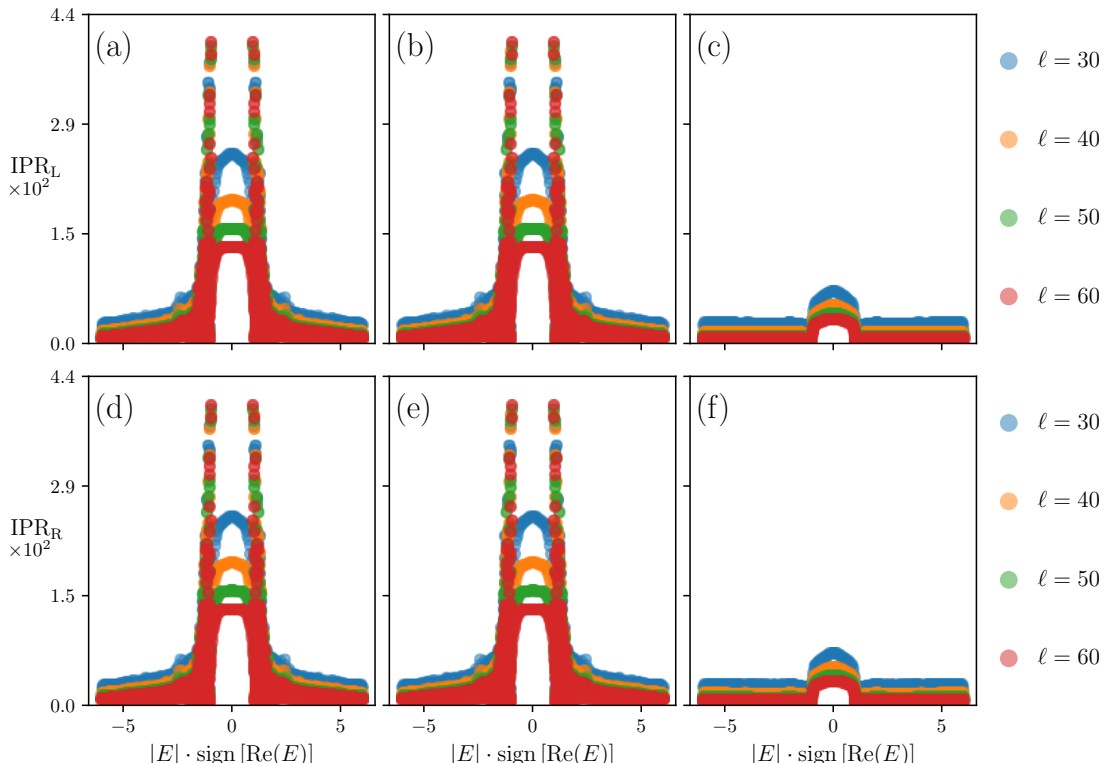

Figure 8: Inverse participation ratio (IPR) analysis for the 2D non-Hermitian Chern insulator from Appendix B. Panels (a), (b), and (c) show IPR spectra for left eigenvectors ($IPR_L$). Panels (d), (e), and (f) show IPR spectra for right eigenvectors ($IPR_R$). The left, center, and right columns correspond to turning on only one non-Hermitian (NH) perturbation $\delta_1$, $\delta_2$, and $\delta_3$, respectively. In (a), (b), (d), and (e) sharp spikes near bulk-band edges show a few states accumulating on one boundary due to the NH skin effect, which is present only for $\delta_1$ and $\delta_2$ perturbations. Such spikes in the IPR spectra is absent in (c) and (f), confirming the absence of NH skin effect for $\delta_3$ perturbation. A broader plateau visible in all the panels indicates weaker localization characteristic of edge states near zero energy. All IPR values consistently decrease as the linear system size $\ell$ increases. Results are obtained for $t = B = 1$, $\Delta_1 = 6$, and $\delta_j = 0.25$ for $j = 1$ (left column) or $2$ (center column) or $3$ (right column).

This model is invariant under the reflection about (a) the $x$ axis, generated $R_x = \tau_3 \mathcal{K}$, under which $(k_x, k_y) \rightarrow (k_x, -k_y)$ and (b) the $y$ axis, generated by $R_y = \mathcal{K}$, under which $(k_x, k_y) \rightarrow (-k_x, k_y)$. Here, $\mathcal{K}$ is the complex conjugation. It is also invariant under four-fold ($C_4$) rotation about the $z$ axis, generated by $R_{\pi/2}^z = \exp[-i\tau_3\pi/4]$ under which $(k_x, k_y) \rightarrow (k_y, -k_x)$. Finally, we note that the model Hamiltonian is also invariant under the inversion, which is equivalent to rotation about $z$ axis by an angle $\pi$, generated by $R_\pi^z = \tau_3$, under which $k \rightarrow -k$. However, in two dimensions, the inversion is the product of $R_x$ and $R_y$.

With the spatial symmetries of $H_{\text{Chern}}^{\text{Her}}$ in hand, we now introduce the following local anti-Hermitian operator to address the resulting NH skin effect

$$H_{\text{AH}}(\delta_1, \delta_2, \delta_3) = i \sum_{j=1}^{3} \delta_j \Gamma_j = \delta_1(i\tau_1) + \delta_2(i\tau_2) + \delta_3(i\tau_3). \tag{B.2}$$

The symmetry transformations of each term under the spatial symmetries of the Hamiltonian $H_{\text{Chern}}^{\text{Her}}$ are shown in Table 1. In particular, the term proportional to $\delta_1$ ($\delta_2$) breaks the reflec-

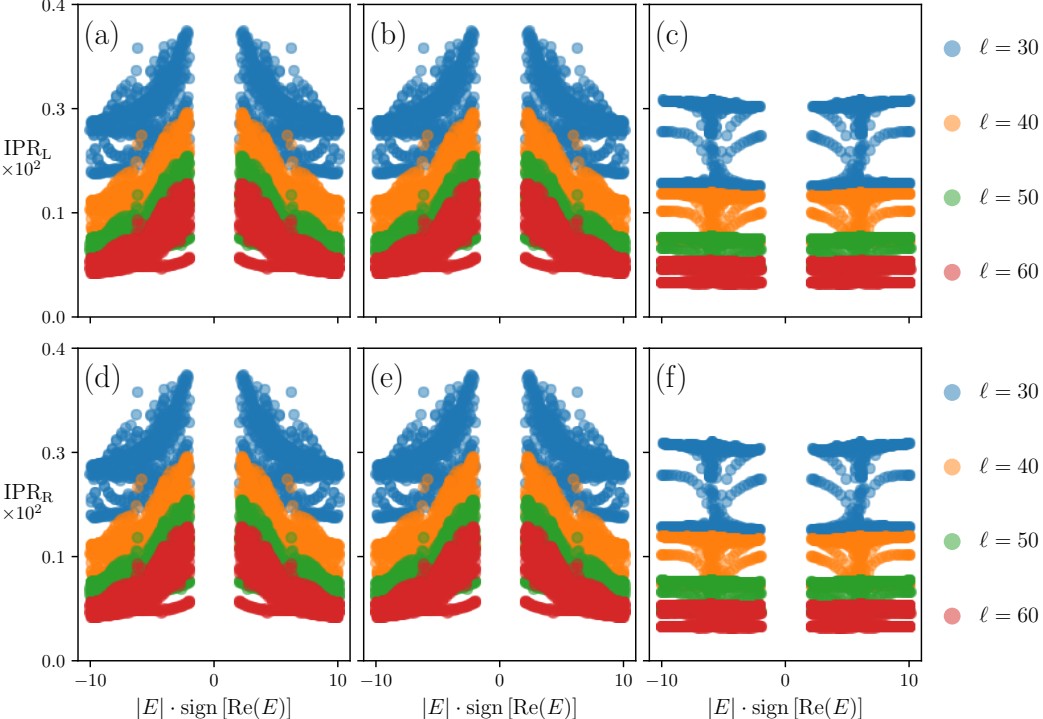

Figure 9: Inverse participation ratio (IPR) analysis for the 2D non-Hermitian insulator model, discussed in Appendix B but with $\Delta_1 = -2$ corresponding to a trivial bulk phase. The panel organization and the rest of the parameters are same as in Fig. 8. Unlike the topological case, no distinct mid-gap plateau emerges, reflecting the absence of topological boundary states in the trivial phase. Instead, the right and left IPR spectra show only broad triangular structures, indicating bulk-state accumulation near spectral edges induced by the NH skin effect (for $\delta_1$ and $\delta_2$). For perturbation $\delta_3$, which does not produce the NH skin effect, the spectra appear appear flat over all $E$. All curves systematically shift downward with increasing linear system size $\ell$.

tion symmetry about $y$ ($x$) axis, while preserving the other reflection symmetry. Consequently, both of them break the inversion symmetry and produce NH skin effect. We also note that these two terms constitute a vector under the four-fold or $C_4$ rotation about the $z$ axis. So, the patterns of the skin effect due to these two terms transform into each other under such a rotation. As $\delta_1$ breaks the reflection about $y$ axis, the skin effect appears along the edges in the $x$ direction, thereby manifesting the reflection symmetry breaking. A similar conclusion also holds for the NH skin effect arising from the $\delta_2$ term. By contrast, the term proportional to $\delta_3$ preserves the inversion symmetry, while transforming as a scalar under four-fold rotation. Hence, it does not show any NH skin effect. These results are shown in Fig. 7. Therefore, from this simple example it is clear that for the appearance of the NH skin effect at least some discrete symmetries must be broken by the anti-Hermitian term, which in $d = 2$ is the inversion symmetry. On the other hand, the reflection symmetries dictate the geometry of the NH skin effects by various anti-Hermitian perturbations and the rotational symmetry ($C_4$ on a square lattice) relates the geometry of the NH skin effects by anti-Hermitian perturbations that together constitute a vector under discrete crystal rotations ($\delta_1$ and $\delta_2$ in this case). In the future, it will be worthwhile extending the symmetry requirement and classification of NH skin effects to other models and in higher dimensions, unfolding the interplay between crystal

symmetries and skin effects. We leave this topic for a future publication. As in our construction, the anti-Hermitian term does not break any crystal or discrete symmetry (for first-order topological phases) or any new discrete symmetry that is not already broken in the Hermitian system (for higher-order topological phases), the NH operator [Eq. (5)] is always guaranteed to be devoid of any NH skin effect.

Existence of the NH skin effect for finite $\delta_1$ and $\delta_2$ perturbations, and its absence with any finite $\delta_3$, are next substantiated from the IPR spectra. All the necessary definitions and diagnostic tools are already discussed in Sec. 4. Therefore, here we only quote the results, shown in Figs. 8 and 9, in the topological and trivial regimes of the model $H_{\text{Chern}}^{\text{Her}} + H_{\text{AH}}(\delta_1, \delta_2, \delta_3)$, respectively. In the topological parameter regime, for perturbations $\delta_1$ and $\delta_2$, the IPR spectra for right- and left-eigenvectors show pronounced spikes at the bulk band edges due to the non-Hermitian skin effect, as bulk eigenstates accumulate against one edge of the lattice. Such NH skin effect-induced spikes are absent for the $\delta_3$ perturbation. Additionally, a flatter region is visible within the bulk gap corresponding to topological edge states for all three NH perturbations, which spread along the entire boundary and thus have lower localization $\sim 1/\ell$, where $\ell$ is the linear dimension of the system in each direction.

Finally, we examine the IPR spectra for the trivial bulk phase, obtained by setting $\Delta_1 = -2$ in the same 2D non-Hermitian model, see Fig. 9. In contrast to the topological case, an IPR plateau is notably absent within the bulk gap for the trivial insulator, confirming the absence of any edge modes. In addition, the perturbations $\delta_1$ and $\delta_2$ lead the right and left eigenvector IPR values to exhibit broad, triangular shapes that peak near the gap rather than forming sharp towers, otherwise observed in the topological regime. This is a consequence of the fact that the complex-band-edge contour does not lie at a single radius, so skin modes occupy a continuous range of $|E|$ rather than occurring at one value. Again, the perturbation $\delta_3$ does not induce the skin effect and so does not show increasing IPR near the gap edge. In all cases, the IPR scales roughly as $1/\ell$, as expected. The IPR analysis shows that the skin modes can coexist with topological boundary modes and extended bulk states.

## C  Topological invariant of NH models: An example

In this Appendix, we show the computation of the topological invariant of the NH operator for a specific case of two-dimensional NH Chern insulator. We begin from the massive Dirac Hamiltonian $H_{\text{Her}}(k)$ [see Eqs. (1)-(4)] with $t_s = \Delta_2 = \Delta_3 = 0$ and $\Gamma_j = \tau_j$ (two-dimensional Pauli matrices) for $j = 1, 2, 3$ therein. Subsequently, from Eq. (5), we can write down the explicit form of the corresponding NH operator to be

$$
\begin{aligned}
H_{\text{Chern}}^{\text{NH}} &= t \left[ \{\sin(k_1 a) - i\alpha \sin(k_2 a)\} \tau_1 + \{\sin(k_2 a) + i\alpha \sin(k_1 a)\} \tau_2 \right] \\
&+ \left[ \Delta_1 - 4B + 2B \{\cos(k_1 a) + \cos(k_2 a)\} \right] \tau_3 \equiv \tau \cdot d(k),
\end{aligned}
\tag{C.1}
$$

where $d(k)$ is a three-component vector and $\tau$ is the vector Pauli matrix. We can then compute the Chern number of this model given by [113]

$$
C = \int_{\text{BZ}} \frac{d^2 k}{4\pi} \left[ \partial_1 \hat{d}(k) \times \partial_2 \hat{d}(k) \right] \cdot \hat{d}(k),
\tag{C.2}
$$

where $\partial_j \equiv \partial_{k_j}$ and $\hat{d}(k) = d(k)/\sqrt{d^2(k)}$, and momentum integral is performed over the first Brillouin zone (BZ). For any $|\alpha| < 1$ (including $\alpha = 0$, corresponding to the Hermitian system), we find $C = +1$ for $0 < \Delta_1/B < 4$ and $C = -1$ for $4 < \Delta_1/B < 8$.

Although, here we explicitly compute the topological invariant for one particular NH model in the following we show that in the parameter regime where all the eigenvalues are guaranteed to be real ($|\alpha| < 1$), all the NH models introduced by following our general principle of construction are connected to a Hermitian model via a similarity transformation. Namely, the NH operator from Eq. (5) can be expressed as

$$H_{\text{NH}}(k, \alpha) = \exp\left[-\Gamma_{d+1}\,\theta/2\right]\left\{\sqrt{1-\alpha^2}\,\left[H_{\text{Dir}}(k) + H_{\text{HOT}}(k)\right] + H_{\text{Wil}}(k)\right\}\,\exp\left[\Gamma_{d+1}\,\theta/2\right],$$
$$\text{(C.3)}$$

but only when $|\alpha| < 1$ (yielding all-real eigenvalues), where $\theta = \tanh^{-1}(\alpha)$. The operator inside the curly parentheses is Hermitian and assumes an identical form as $H_{\text{Her}}(k)$ from Eq. (1), however, in terms of scaled hopping parameter $t \to \sqrt{1-\alpha^2}\,t$ [see Eq. (2)], yielding Dirac kinetic term and magnitude of the discrete symmetry breaking Wilson-Dirac masses $\Delta_j \to \sqrt{1-\alpha^2}\,\Delta_j$ [see Eq. (4)] for $j = 2, 3, \cdots$, responsible for higher-order topology. However, the topological invariant is insensitive to the finite exact magnitude of these parameters and only depends on the ratio $\Delta_1/B$ [see Eq. (3)], appearing in $H_{\text{Wil}}(k)$. Therefore, for $|\alpha| < 1$ when $H_{\text{NH}}(k, \alpha)$ yields all-real eigenvalues, its topological parameter regime, where it hosts robust zero-energy modes and topological invariant therein are identical to a Hermitian operator appearing in Eq. (C.3), which we have explicitly shown for the 2D NH Chern insulator. On the other hand, for $|\alpha| > 1$, the NH operator $H_{\text{NH}}(k, \alpha)$ is related to another NH operator via a similarity transformation

$$H_{\text{NH}}(k, \alpha) = S^{-1}\left\{\sqrt{\alpha^2 - 1}\,\Gamma_{d+1}\left[H_{\text{Dir}}(k) + H_{\text{HOT}}(k)\right] + H_{\text{Wil}}(k)\right\}\,S, \qquad \text{(C.4)}$$

where $\theta = \coth^{-1}(\alpha)$ and $S = \exp\left[\Gamma_{d+1}\,\theta/2\right]$. The operator residing inside the curly parentheses is NH by construction and devoid of any non-trivial topological inavriant.

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
