# Peer review of "Model non-Hermitian topological operators without skin effect: A general principle of construction"

_SciPost Physics Core_

## Round 1 · Referee Report · Anonymous (Referee 2) · 2024-6-10

Report

The authors propose a method to construct the non-Hermitian systems in the manuscrip. The constructed non-Hermitian operator has an energy spectrum that can be only purely real or purely imaginary under periodic boundary conditions (PBC). Given such an energy spectrum, the imaginary magnetic flux is zero [Phys. Rev. B 99, 081103(R)], therefore, the conventional bulk-boundary correspondence is valid and the skin effect is necessarily absent.

This method is ingeniously designed so that the imaginary parts inside the square roots of the eigenvalues either cancel out or do not exist. As a result, the value inside the square root can only be real, and thus the eigenvalues can only be purely real or purely imaginary after taking the square roots.

To conclude, this is an interest work and I would like to recommend its publication.

Requested changes

I would like to list a few minor points and invite the authors to fix them in the final version. 1. For \alpha = 0, the Hermitian Hamiltonian in Figure 1 is H(k) = (sink)\sigma_{x} + (2cosk-1)\sigma_{y}. As far as I know, the form of the Su-Schrieffer-Heeger model is H(k) = (w + vcosk)\sigma_{x} + (vsink)\sigma_{y}. It is clear that the Hamiltonian in Figure 1 is not the Su-Schrieffer-Heeger model.

  1. The energy spectrum in Figure 2(a) is incomplete. According to equation (6), the spectrum should be symmetric around zero.

  2. This method has significant limitations, as it constructs a special class of non-Hermitian models whose energy spectrum is restricted to being purely real or purely imaginary. In general, however, the energy spectrum of non-Hermitian systems is typically complex.

Recommendation

Ask for minor revision

  • validity: good
  • significance: good
  • originality: good
  • clarity: high
  • formatting: excellent
  • grammar: excellent

Author:  Bitan Roy  on 2025-10-15  [id 5938]

(in reply to Report 2 on 2024-06-10)

We thank the referee for the report and supporting the manuscript for publication after some minor changes. Below, we respond to the comments and suggestions from the referee.

1. The referee compactly summarized our key results, giving us confidence regarding the clarity of the presentation. We cite Phys. Rev. B 99, 081103(R) (2019) as Ref. 63.

2. Indeed, the original SSH model is captured by the form proposed by the referee. However, the form of the SSH model we have discussed in Sec. 3.1 is unitarily equivalent to its original version. Hence, these two representations describe the same model and lead to the same topology. We clarify this issue in the revised manuscript.

3. There was a minor error in Fig. 2(a), which gave the impression that the spectrum is not particle-hole symmetric, as spotted by the referee. The x-axis did not encompass all the eigenvalue indices (n), causing this confusion, which we now fix.

4. At no point, did we claim that we are proposing NH topological models that are devoid of any NH skin effect for real, imaginary, and complex eigenvalues. Our goal is to find a class of NH topological model Hamiltonians that are devoid of NH skin effect. And we successfully accomplished this goal, as exemplified by many cases. In the “Discussion & outlook” section, we now mention that in the future it will be worthwhile extending the present discussion to include NH topological models that possess complex eigenvalue spectrum but are still free from NH skin effects.

Changes in the revised manuscript (shown in BLUE):

1. We now cite Phys. Rev. B 99, 081103(R) (2019) as Ref. 63.

2. We state the unitary equivalence between our chosen form of the SSH model and the original one available in the literature (Refs. 82-84).

3. We replace Fig. 2(a) such that the eigenspectrum is particle-hole symmetrical.

4. In “Discussion & outlook” section, we propose an extension of the current work to include NH topological models without the NH skin effect as a future direction.

---

## Round 1 · Referee Report · Anonymous (Referee 1) · 2024-6-10

Strengths

1. The authors have discussed the absence of the non-Hermitian skin effect (NHSE) using a general NH Hamiltonian in arbitrary d-dimensions.
2. Have addressed the issue in dimensions greater than 1.
3. From a single Hamiltonian (Eq[5]) they have constructed different models by changing the parameters.
4. Sufficient illustrations are provided to showcase their results.
5. Have proposed sufficient experimental setups where their observations can be realized

Weaknesses

1. The general principle of construction of NH operators claimed is not rigorously provided.
2. NHSE is a novel phenomenon, therefore NH operators analyzed before the discovery of NHSE did not show this effect. Hence motivation is not clear.
3. Future direction, stated in the discussion and outlook section is not explained properly. The calculation of topological invariants is not a complete research topic and the relevance of this construction for NH superconductors should be elaborated by a couple of sentences.

Report

In their manuscript, the authors address the phenomenon of the non-Hermitian skin effect and illustrate its non-existence in various arbitrary dimensional lattice models. For this purpose, they review a general Bloch Hamiltonian composed of Hamiltonians from different lattice models. They plot the energy eigenvalues and eigenstate to illustrate the absence of NHSE in different models.
They have got good results regarding the NH-SSH model and are consistent with recent literature. They have also addressed this issue in Higher-order topological models by performing the same analysis. They also have calculated the topological invariant for the 2-D Chern insulator.
They have got interesting results for both the eigenenergies and eigenstates of the system with periodic boundary conditions (PBC) as well as open boundary conditions (OBC). The presentation of the results is decent.
Before making any recommendations, I would like to present some major and minor suggestions to the authors, which I kindly request them to address.

Requested changes

Major:

• In the abstract they claim to propose a general principle of construction of NH operators devoid of any skin effect. But in Sec.[3], I find they have defined an NH Hamiltonian using the universal Bloch Hamiltonian, but have not provided any general principle or formalism for construction. I think Sec.[3], should be more comprehensive and elaborate, the authors can clarify why such NH Hamiltonian is devoid of NHSE in terms of symmetry classification in Sec.[3] itself. If different symmetries are responsible for such effects in different dimensions, they can point them out separately (similar to Table[1]) which will convey their findings in a much clearer way.
• In the introduction they wrote ‘typically NH operators display skin effect’ but this sentence is not clear as NHSE is shown only by particular operators. The word ‘typically’ should be clarified.
• In the introduction they also wrote ‘Therefore, construction of NH
topological operators, featuring the BBC in terms of their left or right eigenvectors and thus
generically devoid of the NH skin effect, is of pressing and urgent theoretical and more crucially,
experimental importance.’ People were aware of NH operators without NHSE in photonics and electronics long before even NHSE was discovered. Therefore, the above sentence seems to be misleading. I want to provide some references in support of my point.

1. Non-Hermitian Topological Theory of Finite-Lifetime Quasiparticles: Prediction of Bulk Fermi Arc Due to Exceptional Point. Vladyslav Kozii, Liang Fu.
2. Parity-Time Symmetry meets Photonics: A New Twist in non-Hermitian Optics. Stefano Longhi
3. Parity–time symmetry and exceptional points in photonics. Ş. K. Özdemir et. al.

• Generally, the phenomenon of NHSE is associated with non-reciprocal nature of the underlying Hamiltonian. But recently it has been shown in a 1D NH model that NHSE is absent though the Hamiltonian is non-reciprocal.

Ref. Circuit realization of a two-orbital non-Hermitian tight-binding chain. Dipendu Halder. et. al.

In the above reference, they claim that pseudo-Hermiticity is responsible for suppressing NHSE in their model.
Taking this as a motivation, I would like to suggest the authors emphasize the symmetries (reciprocity, pseudo-Hermiticity, PT) responsible for suppressing NHSE in their models, especially the 3D ones (which is addressed but should be more clearly presented), with the following questions in mind.
1. Is there a general rule on symmetry consideration for NHSE to occur? What symmetries are needed to predict whether I will observe NHSE or not? (For a particular model)
2. In 2D or 3D NH-material is it possible that some states will be extended and some localized due to NHSE? If yes then what constrain should be there in the Hamiltonian?
3. Is it sufficient to look only at the PBC and OBC spectra to comment on NHSE?

I would also propose, if possible, to use non-Bloch analysis in 3D models to mathematically show the non-existence of NHSE.
With this, I hope that the authors can illuminate me as well as strengthen my understanding of NHSE on a much deeper level.

Minor:

• Fig. 1c. All states are exponentially localized at the boundary, why is there no skin effect? I think NHSE that they are trying to convey should be clearly defined, early in the main text.
Ref. Edge States and Topological Invariants of Non-Hermitian Systems. Shunyu Yao1 and Zhong Wang.
• What is the difference between the two inset plots of Fig1c and Fig1d?
• Fig. 2. The black greyscale color code looks odd, I suggest using some light color instead.
• Fig. 3. The black greyscale color code looks odd, I suggest using some light color instead.
• Fig. 4. In the caption it is written ’Panels (c), (f) and (i) are same as (a), (d) and (g), respectively, but for α = 10….’. They are not the same kind of plots, panels (a), (d) and (g) are index vs eigenenergy, whereas panels (c), (f), and (i) are real vs imaginary eigenspectrum plots.
• What is the relevance of Fig 5? Along the skin effect line of thought?
• 3.1 NH topological insulator: One dimension. It is written ‘This model never shows NH skin effect, as anticipated’. Please be clear about this anticipation.
• Discussion and outlook ‘In order to numerically ensure the bi-orthonormality condition … respectively, we sometimes have to add an extremely small amount of random charge disorder (∼ 10−4−10−6).’ They could have shown this explicitly in the appendix as this is important. This phenomenon happens at an Exceptional point (EP), but the manuscript does not mention EPs.
• In Appendix B., and Appendix C. how \Gamma matrices are redefined to \tau matrices should be specified. Moreover, as a whole, I think the form of the \Gamma matrices should be explicitly given.

I cannot recommend the publication of this manuscript in Scipost in the current version, as the motivation is unclear to me. I would strongly suggest the authors rewrite the motivation part and state the novelty of their work clearly. In addition, the definition of the NHSE that is used should be clearly mentioned in the introduction. However, I will rethink my decision if they answer my queries and implement the suggestions provided.

Recommendation

Ask for major revision

  • validity: high
  • significance: good
  • originality: ok
  • clarity: ok
  • formatting: reasonable
  • grammar: good

Author:  Bitan Roy  on 2025-10-15  [id 5939]

(in reply to Report 1 on 2024-06-10)

We thank the referee for the report summarizing our key findings, thus giving us confidence regarding the clarity of the presentation. Below, we respond to the comments/criticisms from the “Strengths”, “Weaknesses” and “Requested changes” section of the report.

1. In response to the comment “Have addressed … greater than 1”, we want to point out that we have addressed the issue in one dimension as well. See Sec. 3.1.

2. We strongly disagree with comment no. 1 from “Weaknesses”. All Altland-Zirnbauer topological phases are described by a massive Dirac Hamiltonian. Upon adding discrete symmetry-breaking Wilson-Dirac masses to it, we realize the higher-order topological (HOT) phases. For all these cases, our proposed “general” principle of construction yields their NH incarnations, devoid of NH skin effects (NHSE).

3. Whether NHSE is a novel phenomenon or not is a matter of perception (comment no. 2 from “weakness”). But it always masks standard bulk-boundary correspondence. Hence, finding NHSE-free NH topological operators, featuring conventional bulk-boundary correspondence, is a subject of pressing importance, which is the motivation of the current work, as stated categorically in the Introduction.

4. In response to comment no. 3 from “weaknesses”, we expand Appendix C on the discussion of a topological invariant. We add a couple of sentences on the future research related to NH superconductors free of NHSE “Discussion & outlook”.

5. We staunchly disagree with the comment “… have not provided any general principle or formalism of construction” from the “Requested changes” section. Instead of repeating our arguments once more, we request the referee to consult our response no. 2 above, which clearly shows that our work presents a general principle of constructing NHSE-free topological models for any Altland-Zirnbauer symmetry class.

The generic absence of NHSE in all our models rests on a simple principle. Either they do not break any crystallographic symmetry (for first-order topological phases) or do not break any new spatial symmetry that was not already broken at the Hermitian level (for HOT phases). Then, in Appendix B, we present an example where due to the lack of spatial symmetry NHSE shows up, to strengthen our claims even further. But tabulating the broken spatial symmetry and the nature of the resulting NHSE in different dimensions have never been the goal or motivation of this work, and it goes far beyond the goal and scope of the present work. This is the subject of a separate investigation.

6. The statement “Typically, NH operators display skin effect …” is self-justified considering the references appearing at the end of this sentence (Refs. 52-79), as almost all the models from those papers show NHSE, following the textbook definition of the word “typical”. We strengthen the statement by mentioning that this statement is true in the context of NH topological phases of matter.

7. We agree that NH operators without NHSE were studied in various contexts before NHSE was discussed in the context of topological phases. However, to the best of our knowledge, prior to our work there was no general principle of constructing NHSE-free model NH operator for topological phases of matter. We nonetheless include the references of the papers that the referee pointed out, discussing NHSE-free NH operators discussed in the context of optics, photonics, and electronics, as Refs. 80 and 81.

8. Although non-reciprocity is a necessary condition for NHSE, it is not sufficient. Notice that some spatial symmetry must be broken to observe NHSE, as exemplified in Appendix C with a specific example. By contrast, our model NH operators for topological phases of matter possess nonreciprocity, but do not display NHSE.

The paper by Halder et. al. PRB 109, 115407 (2024) appeared on arXiv on November 25, 2023, two months after our preprint appeared on arXiv (September 23, 2023). As a matter of fact, their construction of NHSE-free two-orbital model from Sec. C agrees with our general principle of constructing NHSE-free topological model operators, when applied to one dimension. In the revised manuscript, we cite this work as Ref. 94.

In the literature there are many claims on the suppression of NHSE based on reciprocity, pseudo-Hermicity, and PT symmetry. It is impossible for us to review all of them in a research article and comply our constructions with those claims. We presented an unprecedented general principle of constructing NH topological model operators that by virtue of preserving all spatial symmetries or by not breaking any additional symmetry beyond what was already broken at the Hermitian systems, are NHSE free. So, our construction and discussion on NHSE-free topological models are self-contained.

In Appendix C, from an explicit example we have shown that some spatial symmetry must be broken to observe NHSE. As mentioned previously, general symmetry-based criteria for the presence of NHSE in every dimension must be reserved as a subject for a separate investigation and is beyond the goal of the present work.

Although comparing the spectrum and wave-function amplitudes with PBC and OBC gives a good idea regarding the presence or absence of NHSE, it can be more quantitatively established from the inverse participation ratio (IPR), which we now discuss in a new section of the revised manuscript Sec. 4. This analysis shows that even in the presence of NHSE, it is conceivable for some states to be localized and some states to be extended considering the example we discussed in Appendix C.

We fail to appreciate the importance of a “non-Bloch analysis in 3D models to mathematically show the non-existence of NHSE”, given that we have proposed a general symmetry-based construction of NH operators in any dimension that are devoid of NHSE, which we confirm by computing the amplitudes of the wave-functions and now IPR. It is our sincerest opinion that all the claims in the manuscript have been supported and justified by convincing symmetry-based arguments and extensive numerical analysis.

9. All the states are not localized at the boundary in Fig. 1(c), please check the scale of variation, it varies from 1.98 to 2.04. The spikes near the endpoints of the 1D chain are solely due to the topological endpoint modes that are shown in Fig. 1(b). With the new analysis of IPR, detailed in Sec. 4, the absence of NHSE is beyond any doubt.

10. The insets of Figs. 1(c) and 1(d) are for two different parameter values, as the respective corresponding main panels. We fail to see the source of the confusion.

11. The grayscale color code in Fig. 2 and Fig. 3 has been used to facilitate the color-blind readers as much as possible. We request the referee not to micromanage and to spare us from changing it to a light color as the alignment of figures takes a huge amount of time and effort. It is a minor issue with no impact at all.

12. We rewrote the flagged sentence from the caption of Fig. 4.

13. Fig. 5 shows the existence of topological boundary modes in NH topological semimetals as has been previously observed in Hermitian systems, which is one of the main outcomes of this work that in our construction the NH topological phases show the same bulk-boundary correspondence as those in Hermitian systems.

14. The “as anticipated” is based on the general principle of constructing NH topological operators which, by construction, are expected to be devoid of NHSE, which we have discussed in great detail at the beginning of Sec. 3, and then exemplified in Sec. 3.1, 3.2, 3.3, and 3.4 in d=1,2, and 3. We expand the sentence to clarify “as anticipated”.

15. The referee has complicated the statement on the requirement of a small amount of disorder to ensure bi-orthonormality conditions. It has nothing to do with the exceptional points. It is a limitation of Python-based numerical packages for NH systems. When the eigenvalues are too close, the bi-orthonormality condition is not always satisfied numerically. And to ensure this condition we add a small amount of disorder, which we now clarify.

16. In Appendixes B and C, we define the Pauli matrices in terms of Gamma matrices.
The final comment from the referee “motivation is unclear to me” left us completely perplexed. The Title and the Introduction of the paper clearly motivate the present work to find a general principle of constructing NHSE-free operators for both first-and higher-order topological phases. Such construction has never been accomplished (to the best of our knowledge). Admittedly, NH operators without NHSE have been discussed in other fields, such as optics, photonics, and electronics and in the revised manuscript we cite the relevant papers with due credit. But those examples are of no use to construct generic NH topological models without NHSE.

Changes in the revised manuscript (shown in BLUE):

1. We expand Appendix C on the discussion related to the topological invariants for the NH topological models devoid of NHSE that we introduced in this work.

2. We expand the discussion on the future research direction related to NH topological superconductors, devoid of NHSE and add new references as Refs. 98-104.

3. At the end of Appendix B, we state that future work will identify different broken spatial symmetries leading to different types of NHSE.

4. Second paragraph of Introduction: In the sentence starting with “Typically, NH operators display skin effect …” we clarify that this is the case for NH topological models.

5. Second paragraph of Introduction: We state that although NHSE-free model NH operators were studied in various other contexts, such as optics, photonics, and electronics and cite the papers the referee mentioned as Refs. 80 and 81, so far there was no such construction for topological phases, which motivates the current work.

6. Discussion & outlook section: We cite the work by Halder et. al. PRB 109, 115407 (2024) as Ref. 94 and compare their construction with ours.

7. We add a new section on the inverse participation ratio (IPR) as Sec. 4 and Fig. 6 to establish the absence of NHSE in our construction. We have also expanded the discussion from Appendix C with an analysis of the IPR to establish the presence of NHSE and existence of both extended and localized modes in the presence of NHSE. See Figs. 8 and 9.

8. We rewrite the flagged sentence from the caption of Fig. 4 appropriately.

9. We expand the caption of Fig. 5 to clarify its purpose.

10. We further expand the sentence ending with “as anticipated” in Sec. 3.1 to justify the use of this phrase.

11. We clarify the necessity of adding a small amount of disorder to ensure the bi-orthonormality condition while using the Python-based numerical packages for NH systems.

12. In Appendixes B and C, we define the Pauli matrices in terms of the Gamma matrices.

Attachment:

Response_Referees_SciPostCore_NHTopology.pdf

---

## Round 2 · Referee Report · Anonymous (Referee 2) · 2025-10-21

Report

The authors have revised their manuscript according to the previous reports, and I recommend the publication of the revised version.

Recommendation

Publish (easily meets expectations and criteria for this Journal; among top 50%)

---

## Round 2 · List of Changes

Changes in the revised manuscript (shown in BLUE) in Response to Referee 2:

1. We now cite Phys. Rev. B 99, 081103(R) (2019) as Ref. 63.

2. We state the unitary equivalence between our chosen form of the SSH model and the original one available in the literature (Refs. 82-84).

3. We replace Fig. 2(a) such that the eigenspectrum is particle-hole symmetrical.

4. In “Discussion & outlook” section, we propose an extension of the current work to include NH topological models without the NH skin effect as a future direction.

Changes in the revised manuscript (shown in BLUE) in response to Referee 1:

1. We expand Appendix C on the discussion related to the topological invariants for the NH topological models devoid of NHSE that we introduced in this work.

2. We expand the discussion on the future research direction related to NH topological superconductors, devoid of NHSE and add new references as Refs. 98-104.

3. At the end of Appendix B, we state that future work will identify different broken spatial symmetries leading to different types of NHSE.

4. Second paragraph of Introduction: In the sentence starting with “Typically, NH operators display skin effect …” we clarify that this is the case for NH topological models.

5. Second paragraph of Introduction: We state that although NHSE-free model NH operators were studied in various other contexts, such as optics, photonics, and electronics and cite the papers the referee mentioned as Refs. 80 and 81, so far there was no such construction for topological phases, which motivates the current work.

6. Discussion & outlook section: We cite the work by Halder et. al. PRB 109, 115407 (2024) as Ref. 94 and compare their construction with ours.

7. We add a new section on the inverse participation ratio (IPR) as Sec. 4 and Fig. 6 to establish the absence of NHSE in our construction. We have also expanded the discussion from Appendix C with an analysis of the IPR to establish the presence of NHSE and existence of both extended and localized modes in the presence of NHSE. See Figs. 8 and 9.

8. We rewrite the flagged sentence from the caption of Fig. 4 appropriately.

9. We expand the caption of Fig. 5 to clarify its purpose.

10. We further expand the sentence ending with “as anticipated” in Sec. 3.1 to justify the use of this phrase.

11. We clarify the necessity of adding a small amount of disorder to ensure the bi-orthonormality condition while using the Python-based numerical packages for NH systems.

12. In Appendixes B and C, we define the Pauli matrices in terms of the Gamma matrices.

---

## Editorial Decision

in_refereeing